# Impact of Donor Human Milk in an Urban NICU Population

**DOI:** 10.3390/children9111639

**Published:** 2022-10-27

**Authors:** Ahreen Allana, Kahmun Lo, Myra Batool, Ivan Hand

**Affiliations:** 1Department of Pediatrics, NYC Health & Hospitals/Kings County, Brooklyn, NY 11203, USA; 2SUNY-Downstate College of Medicine, Brooklyn, NY 11203, USA

**Keywords:** preterm, breast milk, donor breast milk, necrotizing enterocolitis, neonatal nutrition

## Abstract

The American Academy of Pediatrics recommends the use of donor human milk in infants when mother’s own milk is not available. Our objective was to analyze whether the use of donor human milk in preterm, very-low-birth-weight (VLBW, <1500 g) infants affected the rates of necrotizing enterocolitis, duration of parenteral nutrition (PN), growth, culture-positive sepsis, length of hospital stay, and mortality in an urban NICU population with low exclusive breast-feeding rates. A retrospective cohort study was conducted comparing two 2-year epochs of VLBW neonates before and after the introduction of donor breast milk in our neonatal intensive care unit (NICU). With the introduction of donor human milk, there was a significant reduction in the rate of necrotizing enterocolitis (NEC) (5% vs. 13%; *p* = 0.04) and less severe NEC as defined by Stage III based on the Modified Bell Staging Criteria (10% to 3%; *p* = 0.04). In the donor milk era, there was earlier initiation of enteral feeding (2.69 days vs. 3.84; *p* = 0.006) and a more rapid return to birthweight (9.5 days. 10.9 days; *p* = 0.006). In this study, a change in practice to the use of donor breast milk in a population with low rates of human milk provision was associated with earlier initiation of enteral feeding, faster return to birth weight, and a reduced incidence of NEC.

## 1. Introduction

The American Academy of Pediatrics (AAP) recommends exclusive breastfeeding for the first 6 months of life and has established human milk (HM) as the optimal form of nutrition for infants [1,2]. The use of human milk carries with it many well-documented benefits including easy digestion, growth and maturation factors, immunological development, and allergy prevention [3,4,5,6,7].

Preterm infants represent a particularly vulnerable population because of their increased susceptibility to infection, feeding intolerance, necrotizing enterocolitis (NEC), adverse neurodevelopmental outcomes, and overall morbidity and mortality [8,9]. Studies have shown significant short- and long-term beneficial effects of mother’s own milk in preterm infants including lower rates of sepsis and necrotizing enterocolitis [10,11,12,13,14,15]. This effect has been attributed to the role of mother’s own milk in promoting early colonization of beneficial gut microbiota, thereby contributing to immunological maturation [16,17] as well as direct effects of immune factors in mother’s own milk [18]. The reduction in the incidence of NEC not only implicates decreased morbidity and mortality in the Neonatal Intensive Care Unit (NICU) but also improved long-term growth and neurodevelopmental outcomes [8,19].

The use of human milk in preterm infants has been shown to improve long-term neurodevelopmental outcomes. Data from NICHD report statistically higher mental, motor and behavior scores at 18 and 30 months of age amongst extremely pre-term infants who received human milk [20,21]. Additionally, longitudinal studies on adolescents show higher cognitive scores and greater white matter and total brain volumes amongst subjects who received the greatest proportion of human milk in the NICU [22]. Human milk has also been shown to be associated with a decreased incidence of retinopathy of prematurity (ROP) [23].

The AAP recommends the use of pasteurized donor human milk (DHM) in all preterm infants weighing less than 1500 g if the mother’s own milk is not available or its use is contraindicated [1]. Despite the well-documented benefits of human milk, studies show ethnic and socio-economic discrepancies with regards to both the awareness and acceptance of donor human milk [2,24,25,26,27,28].

In our predominantly African American inner-city population at Kings County Hospital Center (KCHC) in Brooklyn, NY, where more than 95% of the mothers identified themselves as non-Hispanic black, the overall rate of exclusive breastfeeding was found to be <20%, which is significantly lower than the national average. The AAP reported that none of the Healthy People 2020 goals for breastfeeding were met for non-Hispanic black mothers and infants among the 2018 birth cohort [1]. Suboptimal provision of human milk among the non-Hispanic black population has been shown to be associated with a 3.3-times greater incidence of NEC and a 2.2-times increased risk for mortality compared with the non-Hispanic white population [24]. We found that only 34% of our population knew of donor human milk as a feeding option and 62% preferred formula over donor human milk [25]. The maternal education status played a significant role in both the awareness of donor human milk and the acceptance of its use. Interestingly, US-born mothers were more likely to have known about donor human milk than foreign-born mothers but were less acceptant of its use. Mothers born outside the US were more likely to believe that banked human milk was safe for their infants, would not transmit infection, and that its benefits outweighed its risks [25].

It is evident from these findings that despite the unparalleled benefits of human milk in preterm infants, there are significant racial and socio-demographic disparities that limit its use. A literature review has revealed that the major factors contributing to these differences are maternal ethnicity, education status, and income [2,25,26,27,28].

We conducted a retrospective cohort study to determine whether the use of donor human milk affected the rates of NEC, duration of parenteral nutrition (PN) use, growth, culture-positive sepsis (CPS), length of hospital stay (LOS), and mortality in a population with low rates of human-milk provision.

## 2. Materials and Methods

KCHC began using donor human milk as an alternative to formula in January 2016 for preterm infants with a birth weight of less than 1500 g and gestational age <34 weeks. The infants included in this study were admitted to the NICU in KCHC two years before and two years after the introduction of donor human milk, i.e., from January 2014 to December 2015 and January 2017 to December 2018. The transitional year was excluded. Infants with a gestational age (GA) <34 weeks and birthweight <1500 g were eligible for the analysis. Exclusion criteria included the following: admission after 2 days of life, length of stay less than 14 days, major congenital anomalies, or aneuploidy.

Our feeding protocol for infants <1000 g is to start minimal feeds when clinically stable at 10 cc/kg/day, maintain the same feeds for 5 days, and then increase by 15–20 cc/kg/day. Infants between 1000 and 1500 grams are started on feeds of 10 cc/kg/day when clinically stable, increased daily by 15–20 cc/kg/day. We fortify feeds to 24 calories/ounce when the infant receives 60% of the total intake enterally. All infants receive TPN until full feeds are attained at 140 cc/kg.

The primary efficacy endpoints were the days of life (DOL) to reach full feeds (140 cc/kg/day), return to birth weight (RTBW), duration of parenteral nutrition (PN) and weight-gain velocity, length of hospital stay, and mortality. Secondary efficacy endpoints included the incidence and severity of NEC. Tertiary endpoints included the incidence of bronchopulmonary dysplasia (BPD), retinopathy of prematurity (ROP), culture-positive sepsis (CPS), and intraventricular hemorrhage (IVH). We defined NEC by the modified Bell criteria. This classification uses a combination of systemic, intestinal, radiographic, and laboratory findings to grade the severity of NEC and is the most commonly used diagnostic and staging criteria used in clinical practice. It broadly classifies NEC into Stage 1 (suspected), Stage 2 (proven) and Stage 3 (advanced) disease [29].

BPD was defined as the requirement of supplemental oxygen at >36 weeks postmenstrual age. ROP was defined by the ICROP International Classification of Retinopathy of Prematurity) classification. IVH was defined and graded by head ultrasounds using the Papile classification. We calculated the weight-gain velocity as g/kg/day based on a 2-point average between birthweight and discharge (or medical readiness).

Descriptive statistics were used to summarize the data and identify characteristics of the infants in the two groups. T-tests, chi-square, Fischer Exact and Mann–Whitney U tests were used to compare interval, nominal, and ordinal variables. Data are presented as means or counts (percentages). A *p*-value of <0.05 was considered statistically significant. All descriptive and statistical analyses were performed using SPSS 21.

## 3. Results

Clinical and demographic characteristics were similar in the two groups. There were 90 infants in the pre-donor breast milk (DBM) epoch and 100 infants in the post-DBM epoch. The mean gestational age was 27.6 weeks vs. 27.47 weeks (*p* = 0.19), and the mean birth weight was 964 g vs. 982 g (*p* = 0.39) in the pre-DBM and post-DBM cohorts, respectively. Table 1 shows a comparison of nutrition and growth outcomes before and after the introduction of DBM in our NICU.

Figure 1 highlights the differences in primary endpoints between the two groups. In the donor milk era, there was a trend towards early initiation of enteral feeding, which was statistically significant (*p* = 0.006). There was also a faster return to birth weight in infants who received donor human milk (9.5 vs. 10.9 days, *p* = 0.043). There were no significant differences for the attainment of full enteral feeds (21.5 days vs. 24.8 days, *p* = 0.29), daily weight gain (21.29 g/kg vs. 20.25 g/kg, *p* = 0.086), and length of stay (LOS) (61 vs. 66 days; *p* = 0.5) between the post-DBM and pre-DBM groups, respectively. There was a trend towards lower mortality (6% vs. 18%; *p* = 0.065), though these results were also not statistically significant.

Our secondary endpoints included a comparison of the incidence and severity of NEC between the two infant cohorts. After the introduction of donor human milk, there was a significant reduction in the rate of NEC (5% vs. 13%, *p* = 0.04). Figure 2 summarizes the severity of NEC in the infant epochs based on the Modified Bell Staging Criteria.

The incidence of Stage 2 NEC was similar in the two cohorts (3% in the pre-DBM group and 2% in post-DBM group). There was a decline in the proportion of Stage 3 NEC following the introduction of DHM (10% to 3%; *p* = 0.04) (Figure 2).

There was a lower rate of severe IVH in the post-DBP epoch (16% vs. 6%; *p* = 0.02). There were no significant differences in the rates of BPD, ROP, and CPS between the two groups (Table 2).

## 4. Discussion

The beneficial effects of human milk in improving growth outcomes and reducing the NEC and overall mortality are well documented in the literature. However, the vast majority of these studies have been conducted in mixed ethnic cohorts comprising Hispanic, Latino, Native American and non-Hispanic black women. Our study is one of the first to explore the impact of donor human milk in a predominantly (>95%) African American population with traditionally low exclusive breast-feeding rates [25].

The introduction of donor human milk in our NICU led to earlier initiation of enteral nutrition and earlier return to birth weight. This could be explained by the fact that both physicians and nurses were more comfortable starting donor milk feeds earlier as opposed to formula. The earlier return to birth weight in the donor breast milk cohort could be accounted for by the early introduction of feeding and improved feeding tolerance with human milk compared with formula supplementation [17]. Similar findings have been observed in other studies. Assad et. al. reported decreased feeding intolerance (*p* < 0.0001) and a reduction in the number of days to full enteral nutrition (*p* < 0.001) with exclusive human milk feeds [30].

Interestingly, we found no significant difference between the growth velocity in the two infant cohorts (20.35 g/kg/day and 21.29 g/kg/day; *p* = 0.086). This is similar to Assad et. al.’s study wherein the average weight gain was reported to be similar (18.5–20.6 g/day) amongst groups that received human milk only, fortified human milk, a combination of fortified human milk and formula, and formula alone [30]. Similarly, in a retrospective cohort study, Chowning et. al. found equivalent growth outcomes (*p* = 0.17) following the introduction of donor human milk [31]. However, there are conflicting data on growth outcomes in human-milk- versus formula-fed infants, with several studies reporting lower growth velocity and poor weight gain in infants receiving exclusive human milk [15,32]. A meta-analysis of 11 randomized controlled trials comprising 1809 preterm infants demonstrated significantly higher rates of weight gain, linear growth and head growth in formula fed infants [12]. Our feeding protocol includes the fortification of human milk once the infant receives 80 cc/kg of feeds, which may help explain the fact that we saw no difference in growth velocity between human-milk- and formula-fed preterm infants.

The use of donor human milk in our population was associated with a statistically significant decline in the incidence of NEC (*p* = 0.04). We also observed a significant reduction in the incidence of Stage III NEC. Observational and experimental studies from around the world report similar findings [33,34,35,36]. We speculate that with the increased use of donor human milk, less suspected NEC (stage I) was diagnosed due to less feeding intolerance, and less-severe NEC was seen due to protective factors in human milk.

Studies have shown human milk to be associated with a reduction in the length of hospital stay, incidence of sepsis, and overall mortality [10,11,35]. We found no difference in the incidence of culture-positive sepsis in the two infant cohorts. There was a decrease in the length of hospital stay and mortality in the donor milk era, though these results were not statistically significant. This could be because this was a single-center cohort study with a limited sample size. Future large-scale studies involving diversified ethnic groups across multiple centers are needed. We also noted a decrease in the incidence of severe IVH in the donor milk era (*p* = 0.02). We speculate that the reduction could be due to neuroprotective factors in donor breast milk as well as a decrease in NEC severity and morbidity. Similar findings have been noted by Carome et al. after the introduction of an exclusive human milk diet [37].

Another limitation of our study was its retrospective study design. It was not possible to establish the causality and conclusively prove the association between increased exposure to human milk and long-term growth and mortality. We also do not have data on the actual donor milk consent rate, although the provision of donor milk was offered to all mothers once available. There is a need for prospective cohort studies and randomized controlled trials to explore the long-term benefits of human milk in this population subset.

In this retrospective study, a practice change to the use of donor human milk in a population with low rates of human milk provision was associated with early initiation of enteral feeding and faster return to birth weight, as well as a reduced incidence of NEC. Further study in this population is warranted. There is a need for the development of public policy to increase both the access to donor human milk and awareness regarding its beneficial effects, especially amongst populations with low breastfeeding rates [38]. There is also a need to address structural bias and racism to minimize disparities in breastfeeding and ensure equitable health outcomes for all children [39].

## Figures and Tables

**Figure 1 children-09-01639-f001:**
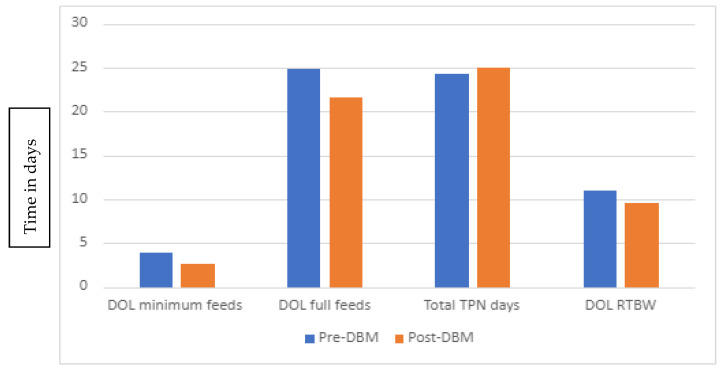
Comparison of enteral, parenteral feeds and return to birth weight (RTBW) between the two infant cohorts.

**Figure 2 children-09-01639-f002:**
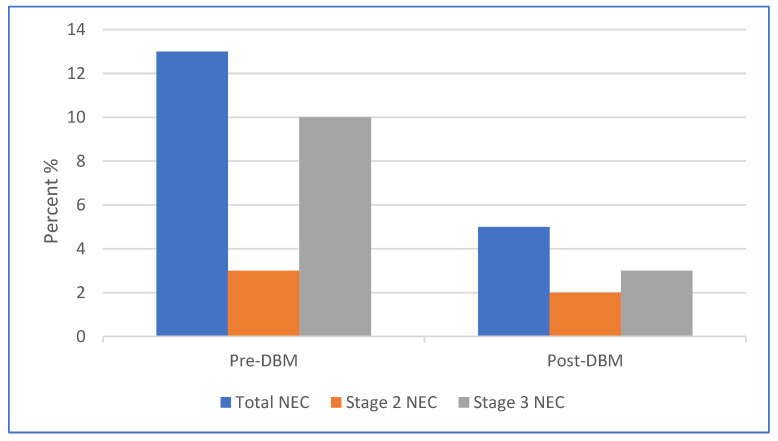
The decrease in NEC and severe NEC between the two infant cohorts.

**Table 1 children-09-01639-t001:** Comparison of mean nutritional and growth outcomes before and after the introduction of donor breast milk.

	Pre-DBM (±SD)	Post-DBM (±SD)	Significance (*p*-Value)
GA (weeks)	27.59 ± 3.2	27.47 ± 5.0	0.19
Birth weight (g)	964 ± 313	982 ± 308	0.39
DOL minimum feeds	3.84 ± 3.3	2.69 ± 2.1	0.006
DOL full feeds	24.88 ± 16.3	21.55 ± 18.9	0.290
Total TPN days	24.30 ± 21.0	24.90 ± 18.4	0.416
RTBW (DOL)	10.92 ± 4.3	9.55 ± 4.8	0.043
Weight gain (g/kg/day)	20.35 ± 6.9	21.29 ± 13.8	0.086

**Table 2 children-09-01639-t002:** Summarizes the differences in tertiary endpoints between the two infant cohorts.

	Pre-DBM (n = 89)	Post-DBM (n = 99)	Significance (*p*-Value)
Severe ROP, %	8 (7)	4 (4)	0.23
BPD, %	19 (17)	28 (28)	0.13
Severe IVH, %	16 (14)	8 (8)	0.02

## Data Availability

Data are available upon reasonable request. The data that support the findings of this study are available on request from the corresponding author. The data are not publicly available due to privacy or ethical restrictions.

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
