# Peer review of "Impact of Donor Human Milk in an Urban NICU Population"

_children, 2022, doi:10.3390/children9111639_

Round 1

Reviewer 1 Report

This was a retrospective study describing the clinical outcomes before and after the availability of donor breast milk in an urban NICU. Specifically, the intent was to report outcomes from an urban setting where there is typically low maternal milk provision, which was generally accomplished. However, the study objective was not novel -- the described outcomes related to the donor milk provision itself, which have been well-described in the literature, and less so to their specific population. Delfosse (J Perinatology 2013) also reported on the implementation of DBM in an urban setting.

Other general comments:

- There is confusion between breastfeeding vs. breast milk provision. Exclusive BF is not possible in the NICU in VLBWs, and BF would obviously occur later in the NICU stay at best. NICU admission alone is a risk factor for low direct breastfeeding rates at discharge.

- Figure 1 and Figure 4 are actually Tables.

- Tables do not report n. Also, the values do not report SD or interquartile range.

- When reporting pre/post differences (e.g. 9.5 vs 10.9 days), there are inconsistencies within the text re: which is reported first -- sometimes it is pre vs. post, sometimes it is post vs. pre.

Remainder of comments by section/lines:

Introduction:

- This section is lengthy. Paragraphs 1-3 could be summarized further, especially since much of it is about human milk in general and not specifically donor milk.

- Line 39-41: While the gut microbiome is affected, presence of immune factors in human milk is also a common explanation for both the decreased incidence of NEC and also sepsis.

- Line 41-44: This is true for mother's milk but not necessarily true for donor milk.

- Line 56-58: Needs citation.

- Line 69-77: This may be better served in the discussion. It detracts from the purpose of the introduction.

Methods:

- Line 96-97: Group sizes are considered results.

- Line 105: ICROP not defined anywhere.

- Line 106: Which method did you use to calculate g/kg/day? 2-point? Average 2-point? Exponential? Also, what are the time points you are using to calculate the velocity?

- Line 108-111: There are more statistical tests than variable qualifiers, so "respectively" is not clear. Also, there are no Pearson correlations reported later on in the Results.

- Line 111: You state that you report medians with quartiles, but in the Results, you refer to mean birth weight. This becomes confusing since the first table does not include IQR or SD, so it is not clear whether you are actually reporting medians or means.

- You mention later on that your institution has a feeding protocol. It may be helpful to describe it generally or to include it in the appendix.

Results:

- Line 122-124: Disagree with this statement (re: days to full feeds), as the result is nowhere near statistically significant.

- Line 127-128: Disagree with this statement (re: length of stay), as the result is nowhere near statistically significant.

- Figure 1 / Table 1: In addition to the comments above, what time point is CGA referring to?

- Figure 2: Why are there more TPN days in the post-DBM era than it takes to get to full feeds?

- Line 141-144: This belongs in the Methods, if included at all.

- Figure 3: It is nearly impossible to distinguish these lines in black & white - need to change to different types of lines (dotted, dashed, etc.). It is also missing a caption.

- Figure 4 / Table 2: There are no % values in the table. Also, later on in the discussion, there is no comment on the difference in severe IVH - could this explain the difference in mortality between the 2 eras? For example, if you control for IVH, does the trend towards difference in mortality disappear?

- What was the donor milk consent rate? How many babies still received formula?

- What was the % maternal milk in the 2 eras? Was there an impact on direct breastfeeding?

Discussion:

Much of your discussion interchanges human milk and donor milk. It would be more powerful to focus your discussion specifically on donor milk literature.

Line 166-168: Were there any other feeding related practice changes that would have impacted the initiation of enteral feeding? It is unusual to wait 3-4 days to start feeding. Does your feeding protocol typically start at day 3? What is your feeding protocol compliance?

Line 168-170: You suggest that there may have been improved feeding tolerance with donor milk in your cohorts, but is that a fair conclusion given that your days to full feeds were not statistically different?

Line 179-181: There are many other papers you could cite on the conflicting growth outcomes on donor milk specifically -- Colaizy (BMC Peds 2012), Fu (Nutrients 2021), O'Connor / DoMINO trial (JAMA 2016).

Line 184-186: How often does your practice fortify higher than 24 kcal/oz?

Line 189-190: More impactful to cite the DoMINO RCT.

Line 203-211: While I agree with the need to increase access to donor milk in VLBWs and the need to address systemic issues that limit maternal milk provision, it is not clear to me that your population's low *maternal milk provision* rate is related to the outcomes of your study. It seems that the conclusion is simply that donor milk provision was associated with the early initiation, faster RTBW, and decreased incidence of NEC.

Author Response

Thank you for your detailed review of our manuscript.  We believe we have addressed all the issues you have raised and have incorporated them into the manuscript with the result being a much more focused work. Point-by-point responses are below:

This was a retrospective study describing the clinical outcomes before and after the availability of donor breast milk in an urban NICU. Specifically, the intent was to report outcomes from an urban setting where there is typically low maternal milk provision, which was generally accomplished. However, the study objective was not novel -- the described outcomes related to the donor milk provision itself, which have been well-described in the literature, and less so to their specific population. Delfosse (J Perinatology 2013) also reported on the implementation of DBM in an urban setting.

Although the objective of our study may not be considered novel, the study of a minority population with less access to donor milk makes it an important population to study. One of the reasons we felt this was important was because in a population where overall breastfeeding rates are low the provision of human milk to preterms is also affected.

Other general comments:

- There is confusion between breastfeeding vs. breast milk provision. Exclusive BF is not possible in the NICU in VLBWs, and BF would obviously occur later in the NICU stay at best. NICU admission alone is a risk factor for low direct breastfeeding rates at discharge.

We have rewritten certain parts of the article relating to the provision of human milk.  Our discussion of exclusive breastfeeding is referring to the overall rate in our total population, not the infants in our NICU. We have tried to organize the paper in such a way as to discuss the breastfeeding issues of our population and then focus on the preterms access to human milk.

- Figure 1 and Figure 4 are actually Tables.  The figures and tables have been relabeled.

- Tables do not report n. Also, the values do not report SD or interquartile range. Values for “n” have been added.  The tables state the results are reported as mean.

- When reporting pre/post differences (e.g. 9.5 vs 10.9 days), there are inconsistencies within the text re: which is reported first -- sometimes it is pre vs. post, sometimes it is post vs. pre.

We have rewritten the differences so the post (intervention) is presented first.

Remainder of comments by section/lines:

Introduction:

- This section is lengthy. Paragraphs 1-3 could be summarized further, especially since much of it is about human milk in general and not specifically donor milk.

Several sentences have been eliminated from the original background.

- Line 39-41: While the gut microbiome is affected, presence of immune factors in human milk is also a common explanation for both the decreased incidence of NEC and also sepsis.

We have added the comment. This effect has been attributed to the role of human milk in promoting early colonization of beneficial gut microbiota, thereby contributing to immunological maturation (16, 17) as well as direct effects of immune factors in human milk.

- Line 41-44: This is true for mother's milk but not necessarily true for donor milk.

That is correct.  There was no reference to donor milk.

- Line 56-58: Needs citation.Citations added 2,24-27

- Line 69-77: This may be better served in the discussion. It detracts from the purpose of the introduction.

Although it could be placed in the discussion we felt poor acceptance  and knowledge of donor human milk in our population was part of the reason for our study so it could be left in the introduction.

Methods:

- Line 96-97: Group sizes are considered results. Group size was moved to results section.

- Line 105: ICROP not defined anywhere.This has now been defined -  ICROP -International Classification of Retinopathy of Prematurity) classification. IVH was defin

- Line 106: Which method did you use to calculate g/kg/day? 2-point? Average 2-point? Exponential? Also, what are the time points you are using to calculate the velocity?

We used 2 point between birthweight and discharge (or medical readiness)

- Line 108-111: There are more statistical tests than variable qualifiers, so "respectively" is not clear. Also, there are no Pearson correlations reported later on in the Results.

This was changed to read - Descriptive statistics were used to summarize the data and identify characteristics of the infants in the two groups. T-tests, , chi-square, Fischer Exact and Mann-Whitney U tests were used to compare interval, nominal, and ordinal variables. - Line 111: You state that you report medians with quartiles, but in the Results, you refer to mean birth weight. This becomes confusing since the first table does not include IQR or SD, so it is not clear whether you are actually reporting medians or means.

Data are presented as means or counts (percentages).

- You mention later on that your institution has a feeding protocol. It may be helpful to describe it generally or to include it in the appendix.

Our feeding protocol for infants <1000grams is to start minimal feeds when clinically stable at 10cc/kg/day,  and then increase by 15-20 cc/kg/day.

Results:

- Line 122-124: Disagree with this statement (re: days to full feeds), as the result is nowhere near statistically significant. Agreed. This statement was confusing and was changed to read :

There was no significant differences for attainment of full enteral feeds (21.5 days vs 24.8 days, p= 0.29), daily weight gain (21.29g/kg vs 20.25 g/kg, p=0.086), and length of stay  (LOS) (61 vs 66 days; p=0.5) between the post-DBM and pre-DBM groups, respectively.

- Line 127-128: Disagree with this statement (re: length of stay), as the result is nowhere near statistically significant.

Agreed. See above.

- Figure 1 / Table 1: In addition to the comments above, what time point is CGA referring to?

This was confusing and removed.

- Figure 2: Why are there more TPN days in the post-DBM era than it takes to get to full feeds? This was due to some outlier values of infants that got restarted on TPN

- Line 141-144: This belongs in the Methods, if included at all. This was removed.

- Figure 3: It is nearly impossible to distinguish these lines in black & white - need to change to different types of lines (dotted, dashed, etc.). It is also missing a caption.

The lines were changed for clarity

- Figure 4 / Table 2: There are no % values in the table. Also, later on in the discussion, there is no comment on the difference in severe IVH - could this explain the difference in mortality between the 2 eras? For example, if you control for IVH, does the trend towards difference in mortality disappear?

Percentages were added to the table as well as values for n.

- What was the donor milk consent rate? How many babies still received formula? We do not have data on consent rates for donor milk.

- What was the % maternal milk in the 2 eras? Was there an impact on direct breastfeeding?

The hospital rate for exclusive breastfeeding during the 2 eras was less than 20%.   

Discussion:

Much of your discussion interchanges human milk and donor milk. It would be more powerful to focus your discussion specifically on donor milk literature.

We eliminated some of the general discussion of human milk.  We wanted to make the reader aware of the advantages of human milk before discussing the use of donor milk when mother’s milk was unavailable.

Line 166-168: Were there any other feeding related practice changes that would have impacted the initiation of enteral feeding? It is unusual to wait 3-4 days to start feeding. Does your feeding protocol typically start at day 3? What is your feeding protocol compliance?

Our feeding protocol for infants <1000grams is to start minimal feeds when clinically stable at 10cc/kg/day,  and then increase by 15-20 cc/kg/day. Our compliance with the protocol is about 75%.

Line 168-170: You suggest that there may have been improved feeding tolerance with donor milk in your cohorts, but is that a fair conclusion given that your days to full feeds were not statistically different?

We speculate that there may have been mproved feeding tolerance may have Although there was no difference

Line 179-181: There are many other papers you could cite on the conflicting growth outcomes on donor milk specifically -- Colaizy (BMC Peds 2012), Fu (Nutrients 2021), O'Connor / DoMINO trial (JAMA 2016).

The DoMINO trial was added to the references.

Line 184-186: How often does your practice fortify higher than 24 kcal/oz? Very Seldom

Line 189-190: More impactful to cite the DoMINO RCT. We have cited the DoMINO RCT.

Line 203-211: While I agree with the need to increase access to donor milk in VLBWs and the need to address systemic issues that limit maternal milk provision, it is not clear to me that your population's low *maternal milk provision* rate is related to the outcomes of your study. It seems that the conclusion is simply that donor milk provision was associated with the early initiation, faster RTBW, and decreased incidence of NEC.

We agree that the low “maternal milk provision” rate is not directly related to the outcomes of the study but believe that populations such as ours that traditionally have low milk provision can achieve better outcomes if donor milk is made available to them.  There has not been enough outreach in these communities to emphasize the benefits of human milk in the population.

Reviewer 2 Report

The authors of the manuscript “Impact of Donor Human Milk in an Urban NICU population” present results from a retro perspective study investigating the influence of human donor milk on the development of preterm neonates.

Subjects of two cohorts were compared in terms of different endpoint including incidence and severity of NEC. Subjects of the first cohort (n=90) did not receive donor milk, subjects of the second cohort (n=100) received human donor milk. Earlier enteral feeding and reduction of NEC is described as positive outcome from donor milk feeding.

Comments on the manuscript:

The authors use a lot of abbreviations that are not common for readers not familiar with clinical pediatrics. The authors should consider implementing a section to explain the abbreviations the reader can refer to.

Authors should describe when feeding of human donor milk was initiated and if it was fed exclusively or if formula was fed supplementary.

Overall, it is a small study with an important outcome but with some limitations. The authors should connect their data more to other published data in that field.

Line 19: Abbr. LOS used first time without explanation

Line 46: delete sentence, repetition from line 43.

Line 59-62: Does the statement refer only to mothers of preterm or overall?

Line 102: delete comma after dysplasia

Line 102: delete culture proven sepsis, CPS already explained in line 85

Line 127: delete length of hospital stay; LOS already explained in Line 86

Figure 1: is Table 1, renumber following figures, accordingly, consider explanation of abbreviation in a footnote

Figure 2: poor style, title of y-axis missing

Figure 3: poor style, title of y-axis missing, description of figure missing, two data points do not allow for drawing a line; overall, better do not use a figure but a table with numbers and statistics

Figure 4: is a table, consider explanation of abbreviation in a footnote

Line 166-168: The authors state that the clinic personal start feeding donor milk earlier compared to formula, that’s interesting but how could that influence the measured outcomes?

Line 187-190: Please explain why Stage 2 NEC is not influenced when Stage 1 and 3 were found to be different in the two cohorts.

5. Conclusions; 6. Patents and follow-on sections: the authors have to fill out or delete the sections correctly

Author Response

Thank you for your review of our manuscript.  We have addressed your concerns below and appreciate your efforts to strengthen our manuscript.

The authors of the manuscript “Impact of Donor Human Milk in an Urban NICU population” present results from a retro perspective study investigating the influence of human donor milk on the development of preterm neonates.

Subjects of two cohorts were compared in terms of different endpoint including incidence and severity of NEC. Subjects of the first cohort (n=90) did not receive donor milk, subjects of the second cohort (n=100) received human donor milk. Earlier enteral feeding and reduction of NEC is described as positive outcome from donor milk feeding.

Comments on the manuscript:

The authors use a lot of abbreviations that are not common for readers not familiar with clinical pediatrics. The authors should consider implementing a section to explain the abbreviations the reader can refer to.

 We have reviewed the manuscript and have added all explanations for abbreviations as they occur.

Authors should describe when feeding of human donor milk was initiated and if it was fed exclusively or if formula was fed supplementary.

We have added a description of our feeding protocol and fortification strategy to the Methods section

Overall, it is a small study with an important outcome but with some limitations. The authors should connect their data more to other published data in that field.

 We have added a reference to the DoMINO trial which also investigated the effects of donor milk. 

Line 19: Abbr. LOS used first time without explanation

 Corrected

Line 46: delete sentence, repetition from line 43.

 Corrected

Line 59-62: Does the statement refer only to mothers of preterm or overall?

 This refers to overall population.

Line 102: delete comma after dysplasia

Corrected

Line 102: delete culture proven sepsis, CPS already explained in line 85

Corrected 

Line 127: delete length of hospital stay; LOS already explained in Line 86

 Corrected

Figure 1: is Table 1, renumber following figures, accordingly, consider explanation of abbreviation in a footnote

 Caption added

Figure 2: poor style, title of y-axis missing

 Corrected

Figure 3: poor style, title of y-axis missing, description of figure missing, two data points do not allow for dra Corrected wing a line; overall, better do not use a figure but a table with numbers and statistics

Figure 4: is a table, consider explanation of abbreviation in a footnote

 Corrected

Line 166-168: The authors state that the clinic personal start feeding donor milk earlier compared to formula, that’s interesting but how could that influence the measured outcomes?

Earlier feeding may have influences earlier return to birthweight.

Line 187-190: Please explain why Stage 2 NEC is not influenced when Stage 1 and 3 were found to be different in the two cohorts.

We speculate that stage 3 NEC which is more severe and progressive was decreased by the protective factors in human milk. The donor milk could not eliminate all NEC but could decrease its severity.

  1. Conclusions; 6. Patents and follow-on sections: the authors have to fill out or delete the sections correctly

These were deleted.

Reviewer 3 Report

The paper presented is an interesting one, but it has - IMHO - some limitations that might be solved by an expansion of both the results and the discussion. Some points:

- in the discussion the authors state that "Our feeding protocol includes fortification of human milk", but there is no trace of this in the methods' section: how much of fortification was used? By excluding the fortification (i.e., grouping the DHM receiving patients in "fortified" and "non-fortified"), is there any difference in growth rates?

- IVH: there is no attempt to hypothesize an explanation for this tertiary endpoint's result

- not all acronyms are esplicited: DOL, CGA... this means that they are not cyted anywhere else in the manuscript. Why? This is particularly important, since DOL corresponds to two of the few significant "p" values.

There are some minor language mistyping, like pre-dominantly (line 59); please check carefully the whole text. Besides, I'm not sure that "racial" is an appropriate word to use, nowadays.

Last point. Sections 5. and 6. should be eliminated in the final version of the manuscript.

Author Response

Thank you for the time spent reviewing our manuscript.  We believe your suggestions have strengthened our report.

The paper presented is an interesting one, but it has - IMHO - some limitations that might be solved by an expansion of both the results and the discussion. Some points:

- in the discussion the authors state that "Our feeding protocol includes fortification of human milk", but there is no trace of this in the methods' section: how much of fortification was used? By excluding the fortification (i.e., grouping the DHM receiving patients in "fortified" and "non-fortified"), is there any difference in growth rates?

We added the following to the Methods section:

Our feeding protocol for infants <1000grams is to start minimal feeds when clinically stable at 10cc/kg/day, maintain the same feeds for 5 days and then increase by 15-20 cc/kg/day. We fortify feeds when the infant receives 60% of the total intake enterally.

- IVH: there is no attempt to hypothesize an explanation for this tertiary endpoint's result

There was no clear reason for the decrease in severe IVH although it could be related to the decrease in Stage 3 NEC.

- not all acronyms are esplicited: DOL, CGA... this means that they are not cyted anywhere else in the manuscript. Why? This is particularly important, since DOL corresponds to two of the few significant "p" values.

All acronyms have been checked and are addressed in the manuscript.

There are some minor language mistyping, like pre-dominantly (line 59); please check carefully the whole text. Besides, I'm not sure that "racial" is an appropriate word to use, nowadays.

We have corrected some of the mistakes. We have used the word “racial” but believe it is appropriate and hope this paper will be a small step in eliminating disparities.

Last point. Sections 5. and 6. should be eliminated in the final version of the manuscript.

These sections were eliminated from the manuscript.

Round 2

Reviewer 1 Report

Replies to the authors are in blue.

Although the objective of our study may not be considered novel, the study of a minority population with less access to donor milk makes it an important population to study. One of the reasons we felt this was important was because in a population where overall breastfeeding rates are low the provision of human milk to preterms is also affected.

I do not disagree with the importance of studying your population, but your findings are consistent with well-established associations with the use of donor milk. Many of the advantages you describe in your background is related to maternal milk exposure, not donor milk. Please do not conflate their benefits. 

In your intro, you state, "there is a dearth of literature on the benefits of donor human milk in populations with traditionally low exclusive breastfeeding rates." Essentially you are describing that you are comparing donor milk more directly against formula, since maternal milk provision is so low. Again, this has already been studied. The medical benefits of donor milk extend to *all* VLBWs, regardless of population. What would be more interesting to see if whether offering donor milk improved or affected maternal milk provision for your population.

Other general comments:

- There is confusion between breastfeeding vs. breast milk provision. Exclusive BF is not possible in the NICU in VLBWs, and BF would obviously occur later in the NICU stay at best. NICU admission alone is a risk factor for low direct breastfeeding rates at discharge.

We have rewritten certain parts of the article relating to the provision of human milk.  Our discussion of exclusive breastfeeding is referring to the overall rate in our total population, not the infants in our NICU. We have tried to organize the paper in such a way as to discuss the breastfeeding issues of our population and then focus on the preterms access to human milk.

Your manuscript continues to interchange breastfeeding and breast milk provision. This needs to continue to be addressed.

- Figure 1 and Figure 4 are actually Tables.  The figures and tables have been relabeled.

- Tables do not report n. Also, the values do not report SD or interquartile range. Values for “n” have been added.  The tables state the results are reported as mean. Your tables still need to report your standard deviations.

- When reporting pre/post differences (e.g. 9.5 vs 10.9 days), there are inconsistencies within the text re: which is reported first -- sometimes it is pre vs. post, sometimes it is post vs. pre.

We have rewritten the differences so the post (intervention) is presented first.

Remainder of comments by section/lines:

Introduction:

- This section is lengthy. Paragraphs 1-3 could be summarized further, especially since much of it is about human milk in general and not specifically donor milk.

Several sentences have been eliminated from the original background.

- Line 39-41: While the gut microbiome is affected, presence of immune factors in human milk is also a common explanation for both the decreased incidence of NEC and also sepsis.

We have added the comment. This effect has been attributed to the role of human milk in promoting early colonization of beneficial gut microbiota, thereby contributing to immunological maturation (16, 17) as well as direct effects of immune factors in human milk. Thank you for adding, but it needs a citation too.

- Line 41-44: This is true for mother's milk but not necessarily true for donor milk.

That is correct.  There was no reference to donor milk. But your implication is that the statement is true for *all* human milk, but it is only true for maternal milk. This needs to be fixed.

- Line 56-58: Needs citation.Citations added 2,24-27

- Line 69-77: This may be better served in the discussion. It detracts from the purpose of the introduction.

Although it could be placed in the discussion we felt poor acceptance  and knowledge of donor human milk in our population was part of the reason for our study so it could be left in the introduction. Understood. Would recommend summarizing your results and taking out p-values. It reads like a discussion and not an introduction.

Methods:

- Line 96-97: Group sizes are considered results. Group size was moved to results section.

- Line 105: ICROP not defined anywhere.This has now been defined -  ICROP -International Classification of Retinopathy of Prematurity) classification. IVH was defin

- Line 106: Which method did you use to calculate g/kg/day? 2-point? Average 2-point? Exponential? Also, what are the time points you are using to calculate the velocity?

We used 2 point between birthweight and discharge (or medical readiness) - This needs to be reported in the methods.

- Line 108-111: There are more statistical tests than variable qualifiers, so "respectively" is not clear. Also, there are no Pearson correlations reported later on in the Results.

This was changed to read - Descriptive statistics were used to summarize the data and identify characteristics of the infants in the two groups. T-tests, , chi-square, Fischer Exact and Mann-Whitney U tests were used to compare interval, nominal, and ordinal variables.

- Line 111: You state that you report medians with quartiles, but in the Results, you refer to mean birth weight. This becomes confusing since the first table does not include IQR or SD, so it is not clear whether you are actually reporting medians or means.

Data are presented as means or counts (percentages). 

- You mention later on that your institution has a feeding protocol. It may be helpful to describe it generally or to include it in the appendix.

Our feeding protocol for infants <1000grams is to start minimal feeds when clinically stable at 10cc/kg/day,  and then increase by 15-20 cc/kg/day. Thank you for including this in your methods. But what about 1000-1500 grams? Your use of donor milk goes up to 1500 grams. What do you fortify to -- 22 or 24 kcal? When you say 60% of enteral feedings, is that different for all babies or is it at a fixed ml/kg? Is goal feeding volume 140 ml/kg/d for all babies? Also, do all of these babies up to 1500g receive TPN?

Results:

- Line 122-124: Disagree with this statement (re: days to full feeds), as the result is nowhere near statistically significant. Agreed. This statement was confusing and was changed to read :

There was no significant differences for attainment of full enteral feeds (21.5 days vs 24.8 days, p= 0.29), daily weight gain (21.29g/kg vs 20.25 g/kg, p=0.086), and length of stay  (LOS) (61 vs 66 days; p=0.5) between the post-DBM and pre-DBM groups, respectively.

- Line 127-128: Disagree with this statement (re: length of stay), as the result is nowhere near statistically significant.

Agreed. See above.

- Figure 1 / Table 1: In addition to the comments above, what time point is CGA referring to?

This was confusing and removed.

- Figure 2: Why are there more TPN days in the post-DBM era than it takes to get to full feeds? This was due to some outlier values of infants that got restarted on TPN.

- Line 141-144: This belongs in the Methods, if included at all. This was removed.

- Figure 3: It is nearly impossible to distinguish these lines in black & white - need to change to different types of lines (dotted, dashed, etc.). It is also missing a caption.

The lines were changed for clarity

- Figure 4 / Table 2: There are no % values in the table. Also, later on in the discussion, there is no comment on the difference in severe IVH - could this explain the difference in mortality between the 2 eras? For example, if you control for IVH, does the trend towards difference in mortality disappear?

Percentages were added to the table as well as values for n. Could you please discuss your IVH findings? Is the trend towards decreased mortality in the post-DBM era related to the decreased incidence of severe IVH? See question above.

- What was the donor milk consent rate? How many babies still received formula? We do not have data on consent rates for donor milk. I realize it is logical to assume that all/many of your babies in the post-DBM era received donor milk, but without this information, technically you have not actually shown that you increased human milk provision/exposure.

- What was the % maternal milk in the 2 eras? Was there an impact on direct breastfeeding?

The hospital rate for exclusive breastfeeding during the 2 eras was less than 20%.   Is this the breastfeeding rate in the nursery or the NICU? Or is this the % of moms in your cohorts who expressed milk in the NICU? And was there a difference between the 2 eras? Please include this in your table if it is available.

Discussion:

Much of your discussion interchanges human milk and donor milk. It would be more powerful to focus your discussion specifically on donor milk literature.

We eliminated some of the general discussion of human milk.  We wanted to make the reader aware of the advantages of human milk before discussing the use of donor milk when mother’s milk was unavailable.

Line 166-168: Were there any other feeding related practice changes that would have impacted the initiation of enteral feeding? It is unusual to wait 3-4 days to start feeding. Does your feeding protocol typically start at day 3? What is your feeding protocol compliance?

Our feeding protocol for infants <1000grams is to start minimal feeds when clinically stable at 10cc/kg/day,  and then increase by 15-20 cc/kg/day. Our compliance with the protocol is about 75%.

Line 168-170: You suggest that there may have been improved feeding tolerance with donor milk in your cohorts, but is that a fair conclusion given that your days to full feeds were not statistically different?

We speculate that there may have been mproved feeding tolerance may have Although there was no difference - could you finish this thought? This reply was incomplete. I'm not convinced you can conclude there is improved feeding tolerance if your days to full feeds were not different.

Line 179-181: There are many other papers you could cite on the conflicting growth outcomes on donor milk specifically -- Colaizy (BMC Peds 2012), Fu (Nutrients 2021), O'Connor / DoMINO trial (JAMA 2016).

The DoMINO trial was added to the references.

Line 184-186: How often does your practice fortify higher than 24 kcal/oz? Very Seldom

Line 189-190: More impactful to cite the DoMINO RCT. We have cited the DoMINO RCT.

Line 203-211: While I agree with the need to increase access to donor milk in VLBWs and the need to address systemic issues that limit maternal milk provision, it is not clear to me that your population's low *maternal milk provision* rate is related to the outcomes of your study. It seems that the conclusion is simply that donor milk provision was associated with the early initiation, faster RTBW, and decreased incidence of NEC.

We agree that the low “maternal milk provision” rate is not directly related to the outcomes of the study but believe that populations such as ours that traditionally have low milk provision can achieve better outcomes if donor milk is made available to them.  There has not been enough outreach in these communities to emphasize the benefits of human milk in the population. But isn't the outcome for all VLBWs better with the availability of donor milk? This benefit is not specific to your population. Also, while you acknowledge that you cannot establish causality, your phrasing sometimes implies causality (e.g. "It also reduced the incidence of NEC...").

Author Response

Replies to the authors are in blue. Author response in BOLD.

Although the objective of our study may not be considered novel, the study of a minority population with less access to donor milk makes it an important population to study. One of the reasons we felt this was important was because in a population where overall breastfeeding rates are low the provision of human milk to preterms is also affected.

I do not disagree with the importance of studying your population, but your findings are consistent with well-established associations with the use of donor milk. Many of the advantages you describe in your background is related to maternal milk exposure, not donor milk. Please do not conflate their benefits. 

In your intro, you state, "there is a dearth of literature on the benefits of donor human milk in populations with traditionally low exclusive breastfeeding rates." Essentially you are describing that you are comparing donor milk more directly against formula, since maternal milk provision is so low. Again, this has already been studied. The medical benefits of donor milk extend to *all* VLBWs, regardless of population. What would be more interesting to see if whether offering donor milk improved or affected maternal milk provision for your population.

We removed the sentence that you raised concerns over.  It now reads:

It is evident from these findings that despite the unparalleled benefits of human milk in preterm infants, there are significant racial and socio-demographic disparities that limit its use. Literature review has revealed that the major factors contributing to these differences are maternal ethnicity, education status and income (2, 24-27).

We conducted a retrospective cohort study to determine whether the use of donor human milk affected the rates of NEC, culture positive sepsis (CPS), growth, duration of parenteral nutrition (PN) use, length of hospital stay (LOS) and overall mortality in a population with low exclusive breastfeeding rates.

Anecdotally, I can say that the use of donor milk has increased the amount of MOM used in our NICU and the proportion of NICU patients being breastfed at discharge.  Unfortunately, we do not have data on amounts of mothers milk provided before and after the intervention

Other general comments:

- There is confusion between breastfeeding vs. breast milk provision. Exclusive BF is not possible in the NICU in VLBWs, and BF would obviously occur later in the NICU stay at best. NICU admission alone is a risk factor for low direct breastfeeding rates at discharge.

We have rewritten certain parts of the article relating to the provision of human milk.  Our discussion of exclusive breastfeeding is referring to the overall rate in our total population, not the infants in our NICU. We have tried to organize the paper in such a way as to discuss the breastfeeding issues of our population and then focus on the preterms access to human milk.

Your manuscript continues to interchange breastfeeding and breast milk provision. This needs to continue to be addressed.

In response to your comments we have gone through the document and made the following changes.

Line 19  low rates of human milk provision ….

Line 28   The use of human milk carries with it many well documented benefits  including easy digestion, growth and maturation…

Line 64  Suboptimal provision of human milk among the non-Hispanic black population has….

Line 86  low rates of human milk provision ….

Line 235 low rates of human milk provision…

There are a few instances where breastfeeding was left when referring to the population as a whole.

- Figure 1 and Figure 4 are actually Tables.  The figures and tables have been relabeled.

- Tables do not report n. Also, the values do not report SD or interquartile range. Values for “n” have been added.  The tables state the results are reported as mean. Your tables still need to report your standard deviations.

Standard deviations have been added to the table.

- When reporting pre/post differences (e.g. 9.5 vs 10.9 days), there are inconsistencies within the text re: which is reported first -- sometimes it is pre vs. post, sometimes it is post vs. pre.

We have rewritten the differences so the post (intervention) is presented first.

Remainder of comments by section/lines:

Introduction:

- This section is lengthy. Paragraphs 1-3 could be summarized further, especially since much of it is about human milk in general and not specifically donor milk.

Several sentences have been eliminated from the original background.

- Line 39-41: While the gut microbiome is affected, presence of immune factors in human milk is also a common explanation for both the decreased incidence of NEC and also sepsis.

We have added the comment. This effect has been attributed to the role of human milk in promoting early colonization of beneficial gut microbiota, thereby contributing to immunological maturation (16, 17) as well as direct effects of immune factors in human milk. Thank you for adding, but it needs a citation too.

The citation was added:

  1. Thai JD., Gregory KE . Bioactive Factors in Human Breast Milk Attenuate Intestinal Inflammation during Early Life. Nutrients. 2020 Feb; 12(2): 581.

- Line 41-44: This is true for mother's milk but not necessarily true for donor milk.

That is correct.  There was no reference to donor milk. But your implication is that the statement is true for *all* human milk, but it is only true for maternal milk. This needs to be fixed.

“Mothers own milk” was substituted for human milk in lines 37, 39 and 41.

- Line 56-58: Needs citation.Citations added 2,24-27

- Line 69-77: This may be better served in the discussion. It detracts from the purpose of the introduction.

Although it could be placed in the discussion we felt poor acceptance  and knowledge of donor human milk in our population was part of the reason for our study so it could be left in the introduction. Understood. Would recommend summarizing your results and taking out p-values. It reads like a discussion and not an introduction.

Paragraph shortened and p values removed.

Methods:

- Line 96-97: Group sizes are considered results. Group size was moved to results section.

- Line 105: ICROP not defined anywhere.This has now been defined -  ICROP -International Classification of Retinopathy of Prematurity) classification. IVH was defin

- Line 106: Which method did you use to calculate g/kg/day? 2-point? Average 2-point? Exponential? Also, what are the time points you are using to calculate the velocity?

We used 2 point between birthweight and discharge (or medical readiness) - This needs to be reported in the methods.

We added … day “based on a 2-point average between birthweight and discharge (or medical readiness)”

- Line 108-111: There are more statistical tests than variable qualifiers, so "respectively" is not clear. Also, there are no Pearson correlations reported later on in the Results.

This was changed to read - Descriptive statistics were used to summarize the data and identify characteristics of the infants in the two groups. T-tests, , chi-square, Fischer Exact and Mann-Whitney U tests were used to compare interval, nominal, and ordinal variables.

- Line 111: You state that you report medians with quartiles, but in the Results, you refer to mean birth weight. This becomes confusing since the first table does not include IQR or SD, so it is not clear whether you are actually reporting medians or means.

Data are presented as means or counts (percentages). 

- You mention later on that your institution has a feeding protocol. It may be helpful to describe it generally or to include it in the appendix.

Our feeding protocol for infants <1000grams is to start minimal feeds when clinically stable at 10cc/kg/day,  and then increase by 15-20 cc/kg/day. Thank you for including this in your methods. But what about 1000-1500 grams? Your use of donor milk goes up to 1500 grams. What do you fortify to -- 22 or 24 kcal? When you say 60% of enteral feedings, is that different for all babies or is it at a fixed ml/kg? Is goal feeding volume 140 ml/kg/d for all babies? Also, do all of these babies up to 1500g receive TPN?

The following was added to the text:  Our feeding protocol for infants <1000grams is to start minimal feeds when clinically stable at 10cc/kg/day, maintain the same feeds for 5 days and then increase by 15-20 cc/kg/day. Infants between 1000-1500 grams are started on feeds of 10cc/kg/day when clinically stable and increased daily by 15-20 cc/kg/day. We fortify feeds to 24 calories/ouncewhen the infant receives 60% of the total intake enterally.All infants receive TPN until full feeds are attained at 140cc/kg.

Results:

- Line 122-124: Disagree with this statement (re: days to full feeds), as the result is nowhere near statistically significant. Agreed. This statement was confusing and was changed to read :

There was no significant differences for attainment of full enteral feeds (21.5 days vs 24.8 days, p= 0.29), daily weight gain (21.29g/kg vs 20.25 g/kg, p=0.086), and length of stay  (LOS) (61 vs 66 days; p=0.5) between the post-DBM and pre-DBM groups, respectively.

- Line 127-128: Disagree with this statement (re: length of stay), as the result is nowhere near statistically significant.

Agreed. See above.

- Figure 1 / Table 1: In addition to the comments above, what time point is CGA referring to?

This was confusing and removed.

- Figure 2: Why are there more TPN days in the post-DBM era than it takes to get to full feeds? This was due to some outlier values of infants that got restarted on TPN.

- Line 141-144: This belongs in the Methods, if included at all. This was removed.

- Figure 3: It is nearly impossible to distinguish these lines in black & white - need to change to different types of lines (dotted, dashed, etc.). It is also missing a caption.

The lines were changed for clarity

- Figure 4 / Table 2: There are no % values in the table. Also, later on in the discussion, there is no comment on the difference in severe IVH - could this explain the difference in mortality between the 2 eras? For example, if you control for IVH, does the trend towards difference in mortality disappear?

Percentages were added to the table as well as values for n. Could you please discuss your IVH findings? Is the trend towards decreased mortality in the post-DBM era related to the decreased incidence of severe IVH? See question above.

We also noted a decrease in the incidence of severe IVH in the donor milk era (p=.02) We speculate that the reduction could be due to neuroprotective factors in donor breast milk as well as a decrease in NEC severity and morbidity. Similar findings have been noted by Carome et al after the introduction of an exclusive human milk diet.(38)

Additional reference:

  1. Carome K, Rahman A, Parvez B. Exclusive human milk diet reduces incidence of severe intraventricular hemorrhage in extremely low birth weight infants. J Perinatol. 2021;41:535-543.

- What was the donor milk consent rate? How many babies still received formula? We do not have data on consent rates for donor milk. I realize it is logical to assume that all/many of your babies in the post-DBM era received donor milk, but without this information, technically you have not actually shown that you increased human milk provision/exposure.

We do not have data on consent rates. We added “ We also do not have data on the actual donor milk consent rate, although provision of donor milk was offered to all mothers once available.”

- What was the % maternal milk in the 2 eras? Was there an impact on direct breastfeeding?

The hospital rate for exclusive breastfeeding during the 2 eras was less than 20%.   Is this the breastfeeding rate in the nursery or the NICU? Or is this the % of moms in your cohorts who expressed milk in the NICU? And was there a difference between the 2 eras? Please include this in your table if it is available.

This is the hospital exclusive rate for the nursery.

Discussion:

Much of your discussion interchanges human milk and donor milk. It would be more powerful to focus your discussion specifically on donor milk literature.

We eliminated some of the general discussion of human milk.  We wanted to make the reader aware of the advantages of human milk before discussing the use of donor milk when mother’s milk was unavailable.

Line 166-168: Were there any other feeding related practice changes that would have impacted the initiation of enteral feeding? It is unusual to wait 3-4 days to start feeding. Does your feeding protocol typically start at day 3? What is your feeding protocol compliance?

Our feeding protocol for infants <1000grams is to start minimal feeds when clinically stable at 10cc/kg/day,  and then increase by 15-20 cc/kg/day. Our compliance with the protocol is about 75%.

Line 168-170: You suggest that there may have been improved feeding tolerance with donor milk in your cohorts, but is that a fair conclusion given that your days to full feeds were not statistically different?

We speculate that there may have been improved feeding tolerance may have Although there was no difference - could you finish this thought? This reply was incomplete. I'm not convinced you can conclude there is improved feeding tolerance if your days to full feeds were not different.

We speculate that physicians and nurses were more comfortable in starting human milk earlier than formula. This may account for the earlier return to birth weight in the donor milk group which occurred during the first 10 days.  The lack of a difference to days of full feeds may reflect the relatively small sample size and wide variability in the group.

Line 179-181: There are many other papers you could cite on the conflicting growth outcomes on donor milk specifically -- Colaizy (BMC Peds 2012), Fu (Nutrients 2021), O'Connor / DoMINO trial (JAMA 2016).

The DoMINO trial was added to the references.

Line 184-186: How often does your practice fortify higher than 24 kcal/oz? Very Seldom

Line 189-190: More impactful to cite the DoMINO RCT. We have cited the DoMINO RCT.

Line 203-211: While I agree with the need to increase access to donor milk in VLBWs and the need to address systemic issues that limit maternal milk provision, it is not clear to me that your population's low *maternal milk provision* rate is related to the outcomes of your study. It seems that the conclusion is simply that donor milk provision was associated with the early initiation, faster RTBW, and decreased incidence of NEC.

We agree that the low “maternal milk provision” rate is not directly related to the outcomes of the study but believe that populations such as ours that traditionally have low milk provision can achieve better outcomes if donor milk is made available to them.  There has not been enough outreach in these communities to emphasize the benefits of human milk in the population. But isn't the outcome for all VLBWs better with the availability of donor milk? This benefit is not specific to your population. Also, while you acknowledge that you cannot establish causality, your phrasing sometimes implies causality (e.g. "It also reduced the incidence of NEC...").

We changed the wording to make it clear this was an association not causation:

In this retrospective study, a practice change to the use of donor breast milk in a population with low rates of human milk provision was associated with  early initiation of enteral feeding and faster return to birth weight, as well as a reduced  incidence of NEC.

Reviewer 2 Report

Thanks for the revised version of the manuscript.

Still, the two figures are of poor style. The legend of the y-axis of Fig. 1 is still missing (is it days?) and in Fig. 2 the title should not appear in the figure and the words "Axis title" have to be removed. Still I do not recomment to draw lines between two data points with no data point justifying the linearity.

Author Response

Reviewer 2 (Report 2):

Thanks for the revised version of the manuscript.

Still, the two figures are of poor style. The legend of the y-axis of Fig. 1 is still missing (is it days?)

and in Fig. 2 the title should not appear in the figure and the words "Axis title" have to be removed.

Still I do not recomment to draw lines between two data points with no data point justifying the

linearity.

Thank you for your observations

We have added time in days to the y axis in Figure 1.

In Figure 2 we have changed the chart type to a bar graph and have eliminated the linearity as recommended by the reviewer. We have also eliminated the title as well as the words “Axis Title”.

Reviewer 3 Report

Dear authors, thank you for replying to some of my observations. Not all of them were addressed, though (see my notes in your authors' reply3).

I have added other observations and suggestions to your revised text. Please check the attached file.

Author Response

The paper presented is an interesting one, but it has - IMHO - some limitations that might be solved by an expansion of both the results and the discussion. Some points:

  • in the discussion the authors state that "Our feeding protocol includes fortification of human milk", but there is no trace of this in the methods' section: how much of fortification was used? By excluding the fortification (i.e., grouping the DHM receiving patients in "fortified" and "non-fortified"), is there any difference in growth rates?
  • We now state our protocol that " we fortify feeds when the infant receives 60% of the total intake enterally". Since all infants received fortification there are no different fortification groups.
  • IVH: there is no attempt to hypothesize an explanation for this tertiary endpoint's result
  • We added this hypothesis to the discussion"

    We also noted a decrease in the incidence of severe IVH in the donor milk era (p=.02) We speculate that the reduction could be due to neuroprotective factors in donor breast milk as well as a decrease in NEC severity and morbidity. Similar findings have been noted by Carome et al after the introduction of an exclusive human milk diet.(38)

    38. Carome K, Rahman A, Parvez B. Exclusive human milk diet reduces incidence of severe intraventricular hemorrhage in extremely low birth weight infants. J Perinatol. 2021;41:535-543.
  •  

Corrections on paper

  • Highlighted text moved from table to results section as outlined .
  • All acronyms have been explained.

There are some minor language mistyping, like pre-dominantly (line 59); please check carefully the whole text. Besides, I'm not sure that "racial" is an appropriate word to use, nowadays.

Corrected. 

Last point. Sections 5. and 6. should be eliminated in the final version of the manuscript.

Eliminated